

# On the interaction of short linear waves with internal solitary waves

Chengzhu Xu[1] and Marek Stastna[1]

[1]Department of Applied Mathematics, University of Waterloo, Waterloo, Ontario N2L 3G1, Canada

*Correspondence to:* Chengzhu Xu (c2xu@uwaterloo.ca)

**Abstract.** We study the interaction of small-scale internal wave packets with a large-scale internal solitary wave using high-resolution direct numerical simulations in two dimensions. A key finding is that for wave packets whose constituent waves are short in comparison to the solitary wave width, the interaction leads to an almost complete destruction of the short waves. For mode-1 short waves in the packet, as the wavelength increases, a cutoff is reached, and for larger wavelengths the waves in the packet are able to maintain their structure after the interaction. This cutoff corresponds to the wavelength at which the phase speed of the short waves upstream of the solitary wave exceeds the maximum current induced by the solitary wave. For mode-2 waves in the packet, however, no corresponding cutoff is found. Analysis based on linear theory suggests that the destruction of short waves occurs primarily due to the velocity shear induced by the solitary wave, which alters the vertical structure of the waves so that significant wave activity is found only above (below) the deformed pycnocline for overtaking (head-on) collisions. The deformation of vertical structure is more significant for waves with a smaller wavelength. Consequently, it is more difficult for these waves to adjust to the new, solitary wave induced background environment. These results suggest that through the interaction with relatively smaller length scale waves, internal solitary waves can provide a means to decrease the power observed in the short wave band in the coastal ocean.

## 1 Introduction

Internal waves are commonly observed in stably stratified fluids such as the Earth's atmosphere and oceans. They exist in a variety of environmental conditions, including those with background shear currents, and on different length and time scales. Interaction between internal waves and other physical processes occurs on a broad scope (Sarkar and Scotti, 2017), resulting in energy exchange between the waves and the background environment. Based on linear wave theory, Cai et al. (2008) studied internal waves in a shear background current, and found that in addition to the velocity shear across the pycnocline, the vertical structure of the horizontal velocity profile also had a significant influence on the evolution of internal waves. The interaction between mode-1 internal tides and mesoscale eddies was examined in Dunphy and Lamb (2014). The authors found that the interaction, essentially the bending of the paths followed by the wave energy, produced hot and cold spots of energy flux. These took the form of beam-like patterns, and resulted in the scattering of energy from the incident mode-1 to modes-2 and higher. The above mentioned studies were not dependent on the presence of boundaries. Motivated by the fact that internal waves have reflection properties that are different from classical Snell's law, Grisouard and Thomas (2015) investigated the interaction





between near-inertial waves and ocean fronts, and found that inertial waves could travel on two distinct characteristics at a front, one flat and one tilted upward, implying the existence of critical reflections from the ocean surface.

Interaction between internal waves of different length scales also occurs naturally (Sun and Pinkel, 2012). When the disparity in length scales between the participating waves is large, the relatively smaller length scale wave essentially plays the role of the disturbance to the "background flow" induced by the relatively larger length scale wave, as they interact with each other. Previous literature has considered wave-wave interaction in a variety of contexts. For example, using ray theory for linearized waves and the principle of wave action conservation, Broutman and Young (1986) studied the interaction of short high-frequency progressive internal waves and long progressive near-inertial waves, and found that there was a net energy transfer from the inertial wave field to the short internal waves. Lamb (1998) investigated the interaction between two mode-1 internal solitary waves (ISWs), and showed that interaction of solitary waves did not correspond to soliton dynamics, since energy exchange was observed and small-amplitude trailing waves of possibly higher modes were generated. More recently, Stastna et al. (2015) examined the interaction between mode-1 and mode-2 internal solitary (solitary-like in cases when the mode-2 wave was breaking) waves, and demonstrated that the interaction yielded a nearly complete disintegration of the relatively smaller mode-2 wave. In particular, the majority of kinetic energy carried by the mode-2 wave was lost and the disturbance to the flow field after the collision no longer had a mode-2 structure. When the length scales of participating waves are similar, Sutherland (2016) found that nonlinear self-interaction might occur, which resulted in energy being transferred to superharmonic disturbances. These disturbances were a superposition of modes such that the amplitude was largest where the change in background buoyancy frequency with depth was largest.

In this work, we study the interaction of small-scale mode-1 internal waves initialized from linear waves with internal solitary waves (ISWs) initialized from the exact Dubreil-Jacotin-Long equation, using high-resolution direct numerical simulations in two dimensions. Internal waves that are short in terms of wavelength compared to the fluid depth are generally less documented in the nonlinear wave literature. In fact, the derivation of the model equations of most weakly nonlinear theories, such as the Korteweg-de Vries equation and its variations, assumes large horizontal scales and thus filters out short waves (Lamb and Yan, 1996). Nevertheless, such waves occupy a non-negligible portion of the Garrett-Munk spectrum of internal waves in the oceans (Thorpe, 2005), and it is important to understand their behavior in order to fully describe internal wave dynamics.

In section 2 we review both linear and nonlinear theories for describing internal wave dynamics. The problem is formulated in section 3. The simulation results are presented in section 4. A key finding is that for waves that are short in comparison to the ISW width, the interaction leads to an almost complete destruction of the short waves, but that for mode-1 short waves there is a cutoff determined by the wavelength of short waves, and waves longer than this cutoff maintain their structure after interaction. We show that this is a key difference from mode-1-mode-2 interaction, which is examined in Stastna et al. (2015). The energy transfer during the interaction is discussed in section 5, and a summary concluding the findings of this study is given in section 6.



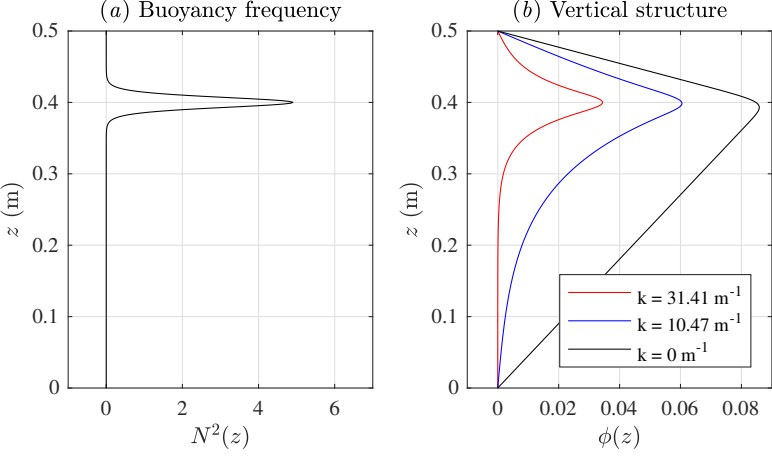

**Figure 1.** Example of (a) buoyancy frequency profile and (b) vertical structure profiles as the wave number varies for mode-1 internal waves in a zero background current. The amplitudes of the eigen-functions have been varied for clarity of visual presentation.

## 2 Internal wave theories

In the classical linear theory, the horizontal structure of internal waves is usually described by the travelling wave ansatz $\exp(ik[x - c_p(k)t])$ where $k$ is the horizontal wave number and $c_p$ is the phase speed, and the vertical structure is described by solutions of the eigenvalue problem often referred to as the Taylor-Goldstein (TG) equation (Kundu et al., 2012). The TG equation is given by

$$\phi_{zz} + \left( \frac{N^2(z)}{(c_p - U)^2} + \frac{U_{zz}}{c_p - U} - k^2 \right) \phi = 0,$$
$$\phi(0) = \phi(H) = 0, \tag{1}$$

where $U = U(z)$ is the background horizontal velocity, $N(z)$ is the buoyancy frequency, and $H$ is the height of the water column. If there are no critical layers (i.e. $c_p - U \neq 0$ for all $z$), for physically relevant $N(z)$, the TG equation has an infinite set of discrete eigenvalues $c_p$ which decrease as $k$ increases and as the mode number increases. The corresponding eigenfunction $\phi(z)$ characterizes the vertical structure of the velocity field (e.g. the wave-induced horizontal velocity is proportional to $\phi_z$). Note that the TG equation simplifies considerably when there is no background shear flow. An example of a single pycnocline buoyancy frequency profile as well as the vertical structure functions for mode-1 waves of particular horizontal wave numbers, in the absence of a shear current, are shown in figure 1. Note that for linear long waves (i.e. waves with $k = 0$) the vertical structure function (and hence horizontal velocity profile) has a non-zero value over the entire water column.

Due to the nonlinear nature of fluid flows, purely linear waves are a mathematical idealization. Especially for large-amplitude waves, results predicted by the linear theory do not agree with measurements. In contrast, internal solitary and solitary-like waves are much more commonly observed in the field (e.g. Klymak and Moum (2003); Scotti and Pineda (2004); also see Helfrich and Melville (2006) for a more complete review). These waves have wave forms different from those predicted by




linear, or even weakly nonlinear, theories (Lamb and Yan, 1996; Lamb, 1999), and are thus often called "fully nonlinear" waves. Fully nonlinear ISWs can be computed by solving the nonlinear eigenvalue problem known as the Dubreil-Jacotin-Long (DJL) equation, which, in a zero background current, takes the form (Stastna and Lamb, 2002; Lamb, 2005)

$$\nabla^2\eta + \frac{N^2(z-\eta)}{c_{\text{isw}}^2}\eta = 0,$$
$$\eta(x,0) = \eta(x,H) = 0,$$
$$\eta(x,z) \to 0 \text{ as } x \to \pm\infty. \tag{2}$$

In this equation, $c_{\text{isw}}$ is the solitary wave propagation speed, and $\eta = \eta(x,z)$ is the vertical displacement of the isopycnal relative to its far-upstream depth. The DJL equation describes translating waves of permanent form and is equivalent to the full set of stratified Euler equations in a frame moving with the wave, where no assumptions are made with respect to the nonlinearity of the fluid flow. Hence, its solutions are exact solitary wave solutions.

## 3    Problem formulation

### 3.1    Governing equations and numerical method

The governing equations for the present work are the incompressible Navier-Stokes equations under the rigid lid and Boussinesq approximations, given by (Kundu et al., 2012)

$$\frac{D\mathbf{u}}{Dt} = -\nabla p - \rho g\hat{k} + \nu\nabla^2\mathbf{u}, \tag{3a}$$

$$\nabla \cdot \mathbf{u} = 0, \tag{3b}$$

$$\frac{D\rho}{Dt} = \kappa\nabla^2\rho, \tag{3c}$$

where $D/Dt$ is the material derivative defined by

$$\frac{D}{Dt} = \frac{\partial}{\partial t} + \mathbf{u} \cdot \nabla. \tag{4}$$

In these equations, $\mathbf{u} = (u, w)$ describes the velocity field with $u$ being the horizontal velocity and $w$ being the vertical velocity, $\rho$ describes the density field (scaled by the reference density $\rho_0$), $p$ is the pressure (again scaled by the reference density $\rho_0$), 20    $g$ is the gravitational acceleration, $\nu$ is the kinematic viscosity, and $\kappa$ is the molecular diffusivity. As is the common practice under the Boussinesq approximation, the equations are in dimensional form, except that the density $\rho$ and pressure $p$ are scaled by the reference density $\rho_0$. For all simulations, we fix the viscosity at $\nu = 10^{-6}$ m$^2$s$^{-1}$ and the diffusivity at $\kappa = 2 \times 10^{-7}$ m$^2$s$^{-1}$. This gives a Schmidt number $Sc = \nu/\kappa = 5$. Periodic boundary conditions are used in the horizontal direction, and free-slip boundary conditions are used at the top and bottom boundaries. We note that no-slip conditions have also been tested 25    but the difference is insignificant, since the majority of the velocity perturbations are found near the pycnocline. The effect of the Earth's rotation is neglected, and hence the simulations are performed in an inertial frame of reference.





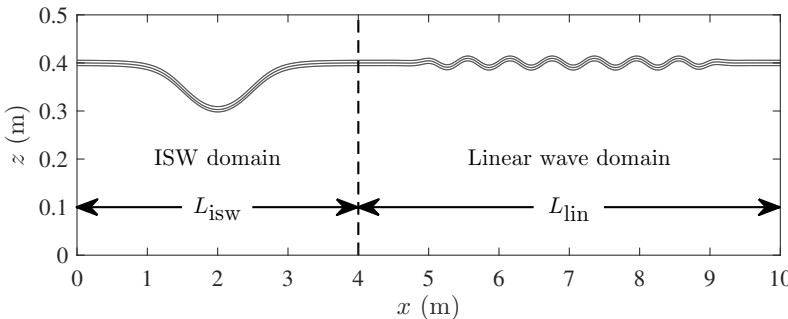

**Figure 2.** Schematic diagram of the model setup. Solid curves are isopycnals indicating the ISW and the linear waves in the initial field.

A complete description of the numerical model used in this study can be found in Subich et al. (2013), where a detailed validation and accuracy analysis through several test cases is also given. The model employs a three-dimensional spectral collocation method, which yields highly accurate results at moderate grid resolutions, though we will employ high resolution in this study in order to resolve the thin pycnocline. For spatial discretization, all simulations performed in this work use

equally spaced grid points in both horizontal and vertical directions. As appropriate for the boundary conditions, the Fourier transform is applied in the $x$-direction, whereas the Fourier sine or cosine transform is applied in the $z$-direction depending on the variable of interest. For time stepping, the model employs an adaptive third-order multistep method, where viscous and diffusive terms are solved implicitly, and pressure is computed via operator splitting.

### 3.2   Model setup and parameter space

A schematic diagram showing the model setup is given in figure 2. The numerical simulations are performed in a two-dimensional, rectangular domain on the laboratory scale, which has an overall length $L_x = 10$ m and a depth $L_z = 0.5$ m. It consists of an ISW subdomain of length $L_{isw} = 4$ m and a linear wave subdomain of length $L_{lin} = 6$ m. The grid size is $N_x \times N_z = 4096 \times 512$, which gives a horizontal grid spacing of 2.44 mm and a vertical grid spacing of 0.98 mm. A right-handed Cartesian coordinate system is considered, where the origin is fixed at the lower left corner of the domain. The $x$-axis

is directed to the rightward along the flat bottom, and the $z$-axis points up towards the surface.

We focus on flows in a quasi two-layer stratification with a dimensionless density difference $\Delta\rho = 0.01$, for which the Boussinesq approximation can be safely adopted. The background density profile, non-dimensionalized by the reference density $\rho_0$, is given by

$$\bar{\rho}(z) = 1 - 0.5\Delta\rho\tanh\left(\frac{z - z_0}{d}\right), \tag{5}$$

where $z_0$ is the location of the pycnocline and $d$ is the half-width of the pycnocline. The specific location of the pycnocline does not affect dynamics of the interaction between the ISW and the linear waves in general, except for the case where the pycnocline is close to the surface such that the ISW could be breaking (Lamb, 2002, 2003). In this work, we set $z_0 = 0.4$ m in order to avoid this situation. The thickness of the pycnocline can affect the gradient Richardson number through the buoyancy





**Table 1.** Solitary wave parameters. Here, the amplitude $\eta_{\max}$ is measured by the maximum isopycnal displacement, and the wavelength $\lambda_{\mathrm{isw}}$ is measured by the half width of the ISW based on the horizontal velocity profile along the inviscid top boundary.

| Amplitude $\eta_{\max}$ (cm) | Propagation speed $c_{\mathrm{isw}}$ (cm s$^{-1}$) | Maximum current $u_{\max}$ (cm s$^{-1}$) | Minimum current $u_{\min}$ (cm s$^{-1}$) | Wavelength $\lambda_{\mathrm{isw}}$ (m) | Reynolds number $Re$ | Richardson number $Ri_{\min}$ |
|---|---|---|---|---|---|---|
| 9.70 | 10.54 | 5.53 | -3.83 | 2.66 | $5.36 \times 10^3$ | 0.45 |

frequency profile it determines, which may have an impact on the interaction. However, this topic is not the focus of the present work (see section 6). In this work, we simply set $d = 0.01$ m for all simulations.

The initial solitary wave is specified by interpolating a solution of the DJL equation onto the ISW subdomain. The DJL equation is solved numerically using the method described in Dunphy et al. (2011). The algorithm for solving the DJL equation specifies the wave parameters implicitly by fixing the available potential energy (Turkington et al., 1991). In this work, we consider the particular solitary wave solution whose parameters are given in table 1. Here, we compute the Reynolds number $Re$ based on the amplitude and maximum wave-induced current:

$$Re = \frac{u_{\max}\eta_{\max}}{\nu}. \tag{6}$$

While there are a variety of Reynolds number estimates available in the literature, this simple estimate is more relevant to the length and velocity scales set by the ISW. The gradient Richardson number $Ri$ is defined by

$$Ri = \frac{N^2}{u_z^2}. \tag{7}$$

It measures the ratio between the strength of the stratification and the shear stress in a parallel shear flow. The minimum Richardson number shown in table 1 is found at the high shear region near the pycnocline along the wave crest. It is much larger than the critical Richardson number 0.25 that provides a necessary (though not sufficient) condition for shear instability in linear theory. Thus, we exclude the possibility of shear instability in our simulations.

We perform a suite of simulations in which the solitary wave propagates to the right and interacts with a small-scale mode-1 wave packet initialized from linear waves. We note that since our numerical model employs the Navier-Stokes equations as the governing equations, these small-scale waves are in fact not purely linear during the simulations. However, for clarity of notation, they will still be referred to as "linear waves", as opposed to the "solitary wave", in the remainder of this paper. The linear waves are specified by solving the TG equation numerically using a pseudo-spectral technique (Trefethen, 2000) in the linear wave subdomain. We consider linear waves of wavelengths ranging from 0.2 to 0.6, whose parameters are given in table 2. Wavelengths out of this range have also been tested. However, for waves with a wavelength less than 0.2 m, self-interaction (Sutherland, 2016) is evident, while for waves with a wavelength larger than 0.6 m, results are similar to cases with a wavelength of 0.6 m (the reason will be discussed later). We examine both overtaking and head-on collisions. An "overtaking collision" means that the ISW and the linear waves propagate in the same direction, whereas a "head-on" collision means that the two propagate in the opposite direction. The amplitude of linear waves is set to be 1 mm for all cases. According





**Table 2.** Linear wave parameters. In the case labels, O indicates an "overtaking" collision and H indicates a "head-on" collision, and the proceeding digits correspond to the wavelength of the linear waves.

| Case label | Wavelength $\lambda$ (m) | Wave number $k$ (m$^{-1}$) | Phase speed $c_p$ (cm s$^{-1}$) | Group speed $c_g$ (cm s$^{-1}$) | Time scale $\tau$ (s) |
|---|---|---|---|---|---|
| | | Cases with an overtaking collision | | | |
| O2 | 0.2 | 31.41 | 3.44 | 1.34 | 110 |
| O2.5 | 0.25 | 25.13 | 3.93 | 1.65 | 113 |
| O3 | 0.3 | 20.94 | 4.36 | 1.97 | 117 |
| O4 | 0.4 | 15.71 | 5.06 | 2.59 | 126 |
| O5 | 0.5 | 12.57 | 5.61 | 3.17 | 136 |
| O6 | 0.6 | 10.47 | 6.05 | 3.69 | 146 |
| | | Cases with a head-on collision | | | |
| H2 | 0.2 | 31.41 | -3.44 | -1.34 | 84 |
| H2.5 | 0.25 | 25.13 | -3.93 | -1.65 | 82 |
| H3 | 0.3 | 20.94 | -4.36 | -1.97 | 80 |
| H4 | 0.4 | 15.71 | -5.06 | -2.59 | 76 |
| H5 | 0.5 | 12.57 | -5.61 | -3.17 | 73 |
| H6 | 0.6 | 10.47 | -6.05 | -3.69 | 70 |

to the linear theory, the propagation of linear internal waves is independent of their amplitude, at least in the limit of small-amplitude waves. In fact, simulations with an amplitude of 2 mm have produced quantitatively similar results, and thus will not be discussed in this paper. In order to ensure a smooth transition across the boundaries between the ISW subdomain and the linear wave subdomain, an envelope function is applied to the amplitude of linear waves. The particular form of the envelope function used here is given by

$$\text{env}(x) = 0.5 \tanh\left[\frac{x - (L_{\text{isw}} + 1)}{0.2}\right] - 0.5 \tanh\left[\frac{x - (L_x - 1)}{0.2}\right], \tag{8}$$

although by testing other forms we found that results are not sensitive to the exact shape of the envelope. For each experiment, we measure the time, $\tau$, over which the solitary wave (which moves at the speed $c_{\text{isw}}$) and the linear wave packet (which moves at the speed $c_g$) experience a full collision cycle by defining

$$\tau = \frac{L_x}{c_{\text{isw}} - c_g}. \tag{9}$$

At $t = \tau$, the location of the solitary wave relative to the linear wave packet is approximately the same as it was in the initial field. In the figures, reported time $T$ is scaled by this quantity such that $T = t/\tau$.

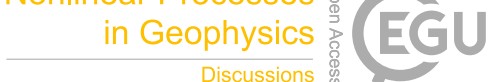

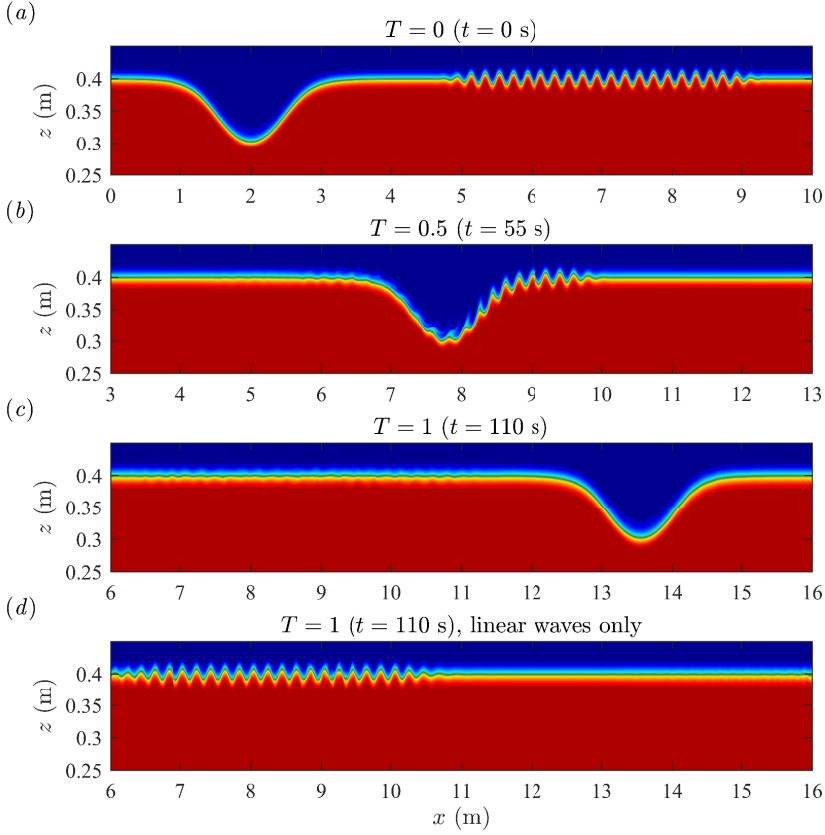

**Figure 3.** Shaded density contours (full range of density shown, green denotes the pycnocline centre) showing the solitary wave and the linear waves in the case O2 (a) before, (b) during and (c) after the collision. Panel (d) shows the corresponding density field from the simulation performed with the same linear wave packet but without the solitary wave. Note the difference in $x$-axis for each panel.

## 4   Simulation results

### 4.1   Evolution of flow fields

An impression of the overall flow behavior in the case O2 can be gained from figure 3. The initial density profile is shown in panel (a), where the disparity in both amplitude and length scale between the solitary wave and the linear waves can be clearly

5    seen. The linear waves have an amplitude that is approximately 10 % of the solitary wave and a wavelength of 7.5 % of the solitary wave. Panel (b) shows that as the linear waves pass through the solitary wave, they are deformed significantly such that they have lost their coherent, wave-like structure almost entirely. Panel (c) shows that after the collision, the disturbance behind the solitary wave has a spatial structure that is completely different from the initial linear waves. To demonstrate that



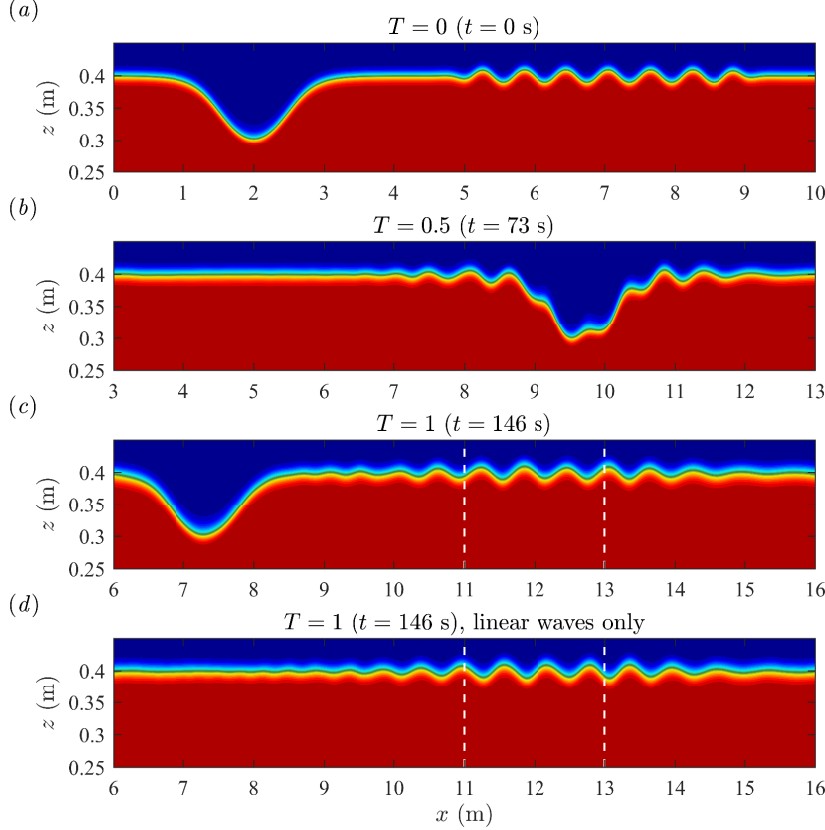

**Figure 4.** Same as figure 3 but for O6. Vertical lines in panels (c) and (d) show the misalignment of wave crests in the two density fields.

such deformation of linear waves does not occur naturally but is a result of the collision, we performed an additional simulation with the same linear wave packet but without the solitary wave. The resulting density field at $T = 1$ is shown in panel (d).

The density profiles of the case O6 are shown in figure 4. The initial density profile, visible in panel (a), shows again the disparity between the solitary wave and the linear waves, though in this case the wavelength of the linear waves is three times larger than that in the previously discussed case (or 26.6 % of the wavelength of the solitary wave). Panels (b) and (c) show, however, that unlike in the previously discussed case, the linear waves are able to retain their spatial structure throughout the collision. The amplitude is also maintained, suggesting that energy loss during the collision is small. Instead, comparison to panel (d), the corresponding density profile obtained from the simulation with linear waves only, suggests that the primary effect of the collision on the linear waves is a phase shift, as indicated by the vertical lines. We will revisit the energy loss in these cases below in section 5.





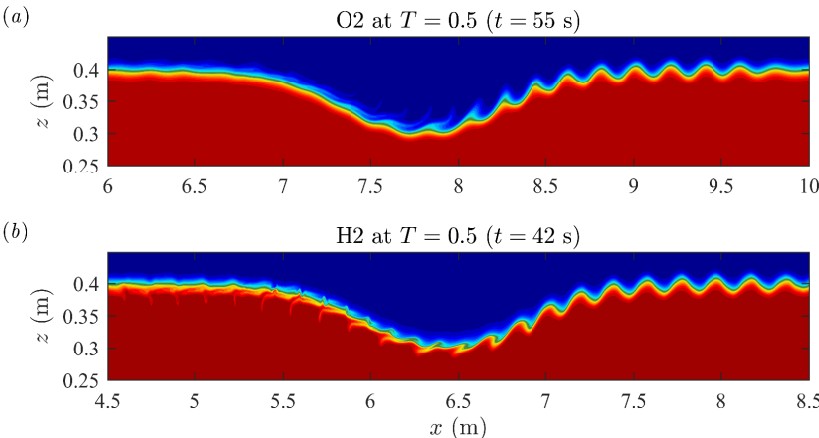

**Figure 5.** Detailed density contours of the simulations (a) O2 and (b) H2, showing the overturning of the linear waves during the collision.

## 4.2 Destruction of short waves

In figure 5, we show the details of the density field during the overtaking collision in case O2; panel (a); and compare it with the density field during the head-on collision in case H2; panel (b). In both cases, the linear waves in front of the solitary wave are unperturbed, whereas those behind the solitary waves are almost completely destroyed. Inside the solitary wave, the

deformation of linear waves in the two cases proceeds in a qualitatively different manner. Panel (a) shows that for the overtaking case, overturning of the linear waves occurs above the pycnocline center, while panel (b) shows that for the head-on collision case, overturning occurs with and below the pycnocline.

To understand what causes the deformation of short waves in these cases, we performed an analysis similar to Stastna et al. (2015), in particular their figure 9. We first extracted the background horizontal velocity and buoyancy frequency profiles at

the crest of the solitary wave.This background state consists of a pycnocline lower than that in an undisturbed situation, and a horizontal velocity with significant shear across the deformed pycnocline. We then computed the linear wave solution with a wavelength of 0.2 m in this background environment using the TG equation (1), and compared it with the solution in the undisturbed background environment. The mode structure functions of these solutions are plotted in figure 6 (a). This figure shows that the vertical structure of the horizontal velocity induced by linear waves is highly dependent on the stratification and

the background current, such that for both overtaking and head-on collision cases, the structure functions at the solitary wave crest (indicated by dashed and dotted-dashed curves) are completely different from their initial, undisturbed state (indicated by a solid curve). The locations of maximum amplitude of the structure functions are shifted downward from their undisturbed situation, in order for the linear waves to adapt to the new, solitary wave-induced background stratification. However, there is a qualitative difference between the overtaking and head on collision as well. Indeed, under the influence of the shear background

current, the structure function in the overtaking (head-on) collision case has its maximum value above (below) the disturbed pycnocline. This is consistent with the observation in figure 5 that inside the solitary wave, perturbations in the overtaking





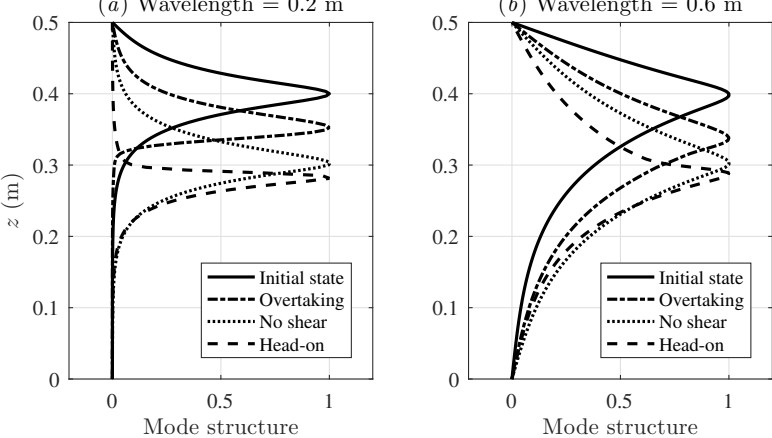

**Figure 6.** Vertical structure profiles of the linear waves with wavelengths of (a) 0.2 m and (b) 0.6 m in the initial, undisturbed state (solid curve) and the ISW-induced background state with an overtaking collision (dotted-dashed curve), a head-on collision (dashed curve) and a hypothetical zero background current (dotted curve). Dotted horizontal lines indicate the location of undisturbed pycnocline center.

(head-on) collision case have a wave-like structure above (below) the pycnocline. We also note that if there is no velocity shear in the background, the vertical structure of linear waves of a given wavelength (e.g. $\lambda = 0.2$ m in this case, as indicated by a dotted curve) depends only on the stratification, regardless of the direction they propagate. Moreover, the vertical structure with respect to the pycnocline center is essentially unchanged. This suggests that the velocity shear in the background alters

the vertical structure of the short waves in a nonlinear manner and leads to the observation that a head-on collision manifests differently from an overtaking collision.

In figure 6 (b), we made the same plot for linear waves with a wavelength of 0.6 m. The figure shows that the key difference in the initial structure function is that it has a non-negligible value over a much larger vertical extend. As a result, at some given depth, in particular that near the deformed pycnocline center, changes of the vertical structure functions from their initial

state are much less dramatic in both overtaking and head-on collision cases. Therefore, longer waves are able to adapt to the ISW-induced background environment more easily and hence are more likely to survive the collision with the solitary wave. We also note that formally changing the amplitude of linear waves does not change their vertical structures and thus does not affect the dynamics of the collision process, though in practice larger amplitude waves are expected to have a different (i.e. Stokes wave) structure.

## 4.3 Comparison to mode-1-mode-2 collision

The above analysis suggests that as the linear waves enter into the solitary wave-induced background state, they are subject to a modified stratification and a velocity shear due to solitary wave-induced current, and it is this velocity shear across the deformed pycnocline that leads to the deformation of short waves. This process is in many ways similar to that found in Stastna et al. (2015). However, a key difference is that the disintegration of mode-2 waves due to the collision is much less dependent



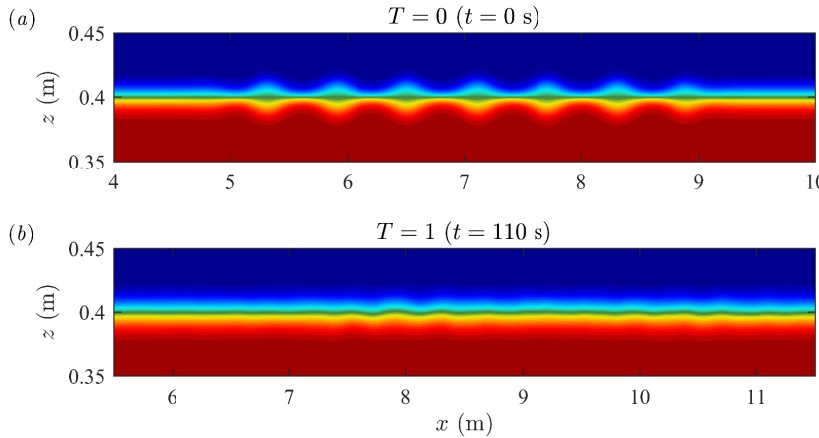

**Figure 7.** Detailed density contours showing the mode-2 wave packet (a) before and (b) after the collision with the solitary wave.

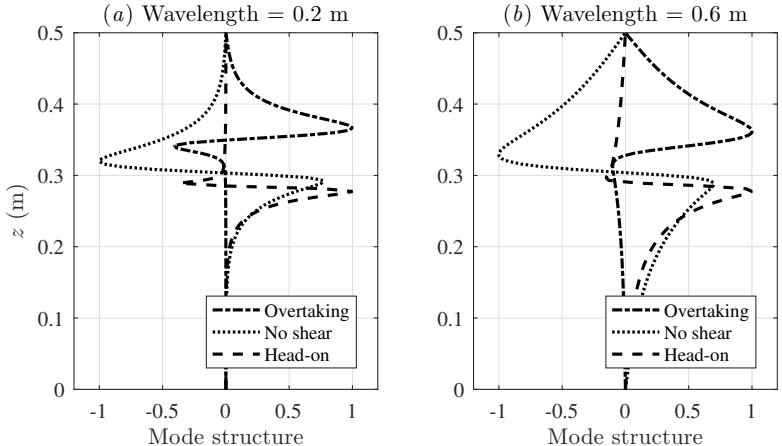

**Figure 8.** Same as figure 6 but for mode-2 waves.

on their wavelength. To compare and contrast with their results, we performed an additional simulation, with mode-2 waves of wavelength $\lambda = 0.6$ m interacting with the same ISW with an overtaking collision. The phase and group speeds of the mode-2 waves are $c_p = 1.43$ cm s$^{-1}$ and $c_g = 1.35$ cm s$^{-1}$, respectively, much smaller than their mode-1 counterparts. Figure 7 shows that after the collision with the ISW, the mode-2 waves are almost completely destroyed, except for some mode-1 like disturbances found near $x = 8$ m in panel (b).

In figure 8, we plotted the vertical structure functions for mode-2 waves in the ISW-induced background environment. The figure shows that the presence of velocity shear leads to significant changes in the vertical structures of horizontal velocity profiles of mode-2 waves with wavelengths of both 0.2 and 0.6 m. In the latter case, the deformed vertical structure functions show characteristics of mode-1 waves, with essentially no perturbation below (above) the pycnocline for the overtaking (head-





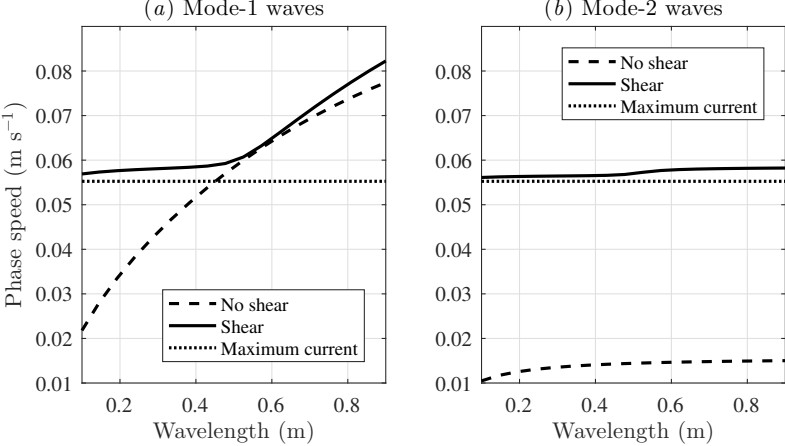

**Figure 9.** Phase speed of (a) mode-1 and (b) mode-2 waves in the ISW-induced background shear current (solid curves) and a hypothetical zero background current (dashed curves), as a function of wavelength. Dotted lines indicate the maximum ISW-induced current.

on) case. This is similar to figure 9 (b) in Stastna et al. (2015), but fundamentally different from our figure 6 (b), implying that mode-1-mode-2 collisions are different from mode-1-mode-1 collisions. In fact, mode-2 waves were unable to maintain their coherent structure after the collision with mode-1 waves in all simulations in Stastna et al. (2015). Recent experiments (M. Carr, personal communication) suggest that the situation is more complex when the mode-1 wave amplitude is comparable to the mode-2 wave amplitude, though it is unclear if such a situation had relevance to situations in the ocean.

## 4.4 Change of phase speed

Recall from figure 4 that a secondary effect of the interaction is a phase shift of the linear waves. To explain this observation, consider the linear long wave speed $c_{\mathrm{lw}}$ in a two-layer stratification, defined by

$$c_{\mathrm{lw}} = \sqrt{\Delta \rho g \frac{h_1 h_2}{H}}, \tag{10}$$

where $h_1$ is the upper layer depth, $h_2$ is the lower layer depth and $H$ is the total depth. The long wave speed sets the limit of the phase speed of linear waves in a two-layer stratification such that $c_p$ approaches $c_{\mathrm{lw}}$ as the wavelength approaches infinity. Thus $c_{\mathrm{lw}}$ provides a good estimate of the maximum phase speed in a quasi-two layer stratification. Using the long wave speed as a guide, we note that the phase speed reaches its maximum value when $h_1 = h_2$ (i.e. when the two layers are equal in depth), provided other parameters (e.g. wavelength) remain constant. In our simulations, since we consider an ISW of depression, the pycnocline at the wave crest is close to the mid-depth. Hence, the linear waves will experience an increase in phase speed as they propagate through the ISW.

In addition to the stratification, the presence of background current will also modify the phase speed. In figure 9, we explore the change of phase speed due to the presence of ISW-induced shear current for both mode-1 and mode-2 waves. For mode-1 waves shown in panel (a), in the long wave limit, the phase speed in the shear background current is very close to that in



a zero background current. However, in the short wave limit, the figure shows that the phase speed in the shear background current approaches the maximum ISW-induced current, whereas the phase speed in a zero background current approaches zero instead. This again suggests that short mode-1 waves are more likely to be influenced by the nonlinear interaction with ISW. In particular, the critical wavelength that determines whether the phase speed is significantly influenced by the shear current

is approximately 0.5 m, where the phase speed in a zero background current intersects the maximum velocity of the shear current. On the other hand, for mode-2 waves shown in panel (b), the phase speed is altered by the shear current throughout the whole spectrum of wavelength, since the phase speed in a zero background current is less dependent on the wavelength and is always much smaller than the maximum velocity of the shear current. This is also consistent with the fact that mode-2 waves are less persistent after nonlinear interactions with the ISW. The fact that the ISW-induced maximum current essentially sets

the lower limit for the phase speed of short waves implies that a critical layer does not exist for the ISW used in our simulations (as well as those with smaller amplitude). While the above analysis is performed for overtaking collisions (i.e. for linear waves propagating to the right), we also examined head-on collisions (not shown) and found similar behavior in the phase speed versus wavelength plots.

## 5   Energetics

### 5.1   Diagnostic tool: Power spectral density

A function that describes a physical process can be represented either in the physical space or in the Fourier space. The two different representations are connected through the Fourier transform. Suppose $f$ is a function of position $x$ in the physical space, then the corresponding Fourier transformed variable $F$ is a function of the horizontal wave number $k$ and is given by

$$F(k) = \int_{-\infty}^{\infty} f(x)e^{ikx}dx. \tag{11}$$

If $x$ is bounded, then $k$ takes discrete values

$$k = k_n = \frac{2n\pi}{L_x}, \quad n = 0, 1, \ldots, \infty. \tag{12}$$

Parseval's theorem states that the total power in a signal is the same whether it is computed in the physical space or in the Fourier space (Press et al., 2002). That is,

$$\text{Total Power} = \int_{0}^{L_x} |f(x)|^2 dx = \sum_{n=0}^{\infty} |F(k_n)|^2 dk. \tag{13}$$

From this theorem, we can define the power spectral density (PSD) of the function $f$ as

$$\text{PSD} = |F(k)|^2. \tag{14}$$

The PSD is a function of the wave number $k$. It can be interpreted as the strength of the signal at each wave number. For this reason, it provides a powerful tool for analyzing physical processes.

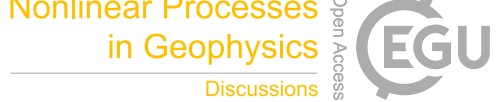



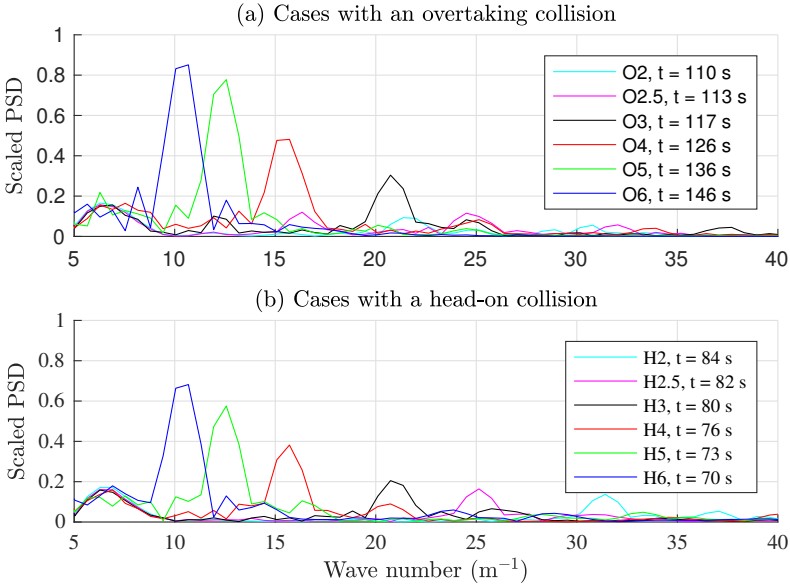

**Figure 10.** Scaled PSD of linear waves in the simulations with (a) an overtaking collision and (b) a head-on collision at $T = 1$.

In the remainder of this section, we compute the PSD of horizontal velocity in the layer above the pycnocline and use it to estimate the amount of wave energy being transferred during the collisions. According to the linear wave theory, the wave-induced horizontal velocity is much larger than the vertical velocity and hence is the dominant contribution to the kinetic energy. The location of horizontal layer chosen for the analysis is $z = 0.43$ m (i.e. 3 cm above the pycnocline), though we

have also calculated the PSD at other depths and found that results are not sensitive to the particular choice of horizontal layer. We restrict our attention to the portion of wavenumber space between 5 and 40 m$^{-1}$, for the reason that wavelengths being examined in our simulations are between 0.2 m and 0.6 m. The PSD is computed at scaled times $T = 1$ and scaled by the peak values observed at $T = 0$, in order to quantify the change of wave energy due to the interaction. According to Parseval's theorem, this quantity remains the same when mapped back into the physical space. Thus, it represents the relative strength of

horizontal current at $T = 1$ and hence provides an estimate of the percentage of kinetic energy remaining after one full collision cycle.

## 5.2 Energy exchange due to the interaction

In figure 10 we examine the energy transfer of linear waves due to collision by plotting the PSD of horizontal velocity in the wave number domain. The figure shows that in all cases, there is a net loss of wave energy during the collision. It also

shows that for a given solitary wave, the wavelength of the linear waves (which remains unchanged after the collision) is the single most important factor that determines the amount of PSD (and hence wave energy) remaining after the collision. While the longest waves may retain as much as 85% of the kinetic energy they had initially, the shortest waves lose almost





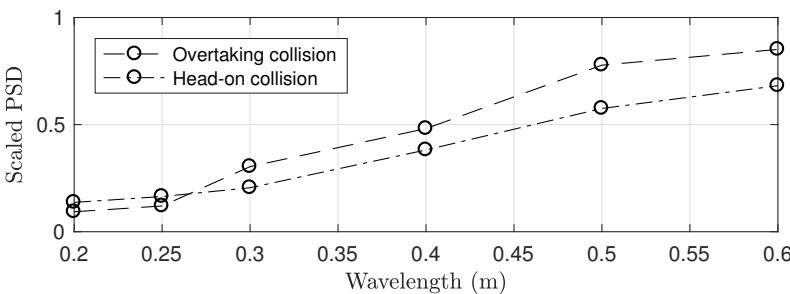

**Figure 11.** Maximum values of the scaled PSD observed in figure 10 versus their corresponding wavelengths.

**Table 3.** Quantitative measurement of the maximum values plotted in figure 11.

| Wavelength | Simulation and scaled PSD | | | |
|---|---|---|---|---|
| | Overtaking collision | | Head-on collision | |
| 0.2 m | O2 | 9.08 % | H2 | 13.69 % |
| 0.25 m | O2.5 | 15.69 % | H2.5 | 16.43 % |
| 0.3 m | O3 | 30.39 % | H3 | 20.52 % |
| 0.4 m | O4 | 48.14 % | H4 | 38.17 % |
| 0.5 m | O5 | 77.76 % | H5 | 57.54 % |
| 0.6 m | O6 | 85.11 % | H6 | 68.18 % |

all of their initial energy such that the peaks of the PSD can hardly be distinguished from background noise. Among other factors, a head-on collision is slightly more efficient in destroying the linear waves than an overtaking collision, except for the small wavelength limit. This may be explained by the fact that during a head-on collision, the structure function (especially its peak) shifts further away from its initial state than during an overtaking collision, as shown in figure 6, such that the new,

ISW-induced background environment is more difficult for the linear waves to adjust to. In contrast, the initial amplitude of the linear waves has very little impact on the net energy transfer due to collision, since curves produced from simulations in which the linear waves have an amplitude of 0.2 m (not shown) are almost exactly the same as their counterparts shown in panel (a). Though we did not consider large amplitude short waves, since these will have their own complex dynamics.

The maximum value of the scaled PSD as a function of wavelength is plotted in figure 11, along with a quantitative mea-
surement in terms of percentage given in table 3. The figure and table show that the maximum value of the scale PSD increases monotonically as the wavelength increases, for both overtaking and head-on collisions. It approaches zero in the short wave limit and one in the long wave limit. For waves with a wavelength much longer than 0.6 m, simulation results (not shown) suggest that the maximum values of the scale PSD at $T = 1$ are at the level of 90 % but are never larger than 100 %, implying that very little wave energy is being transferred from the short waves during the collision and that no energy is transferred from
the ISW to the short waves. For the longest waves, the slight decrease in PSD is at a similar level to viscous dissipation. We





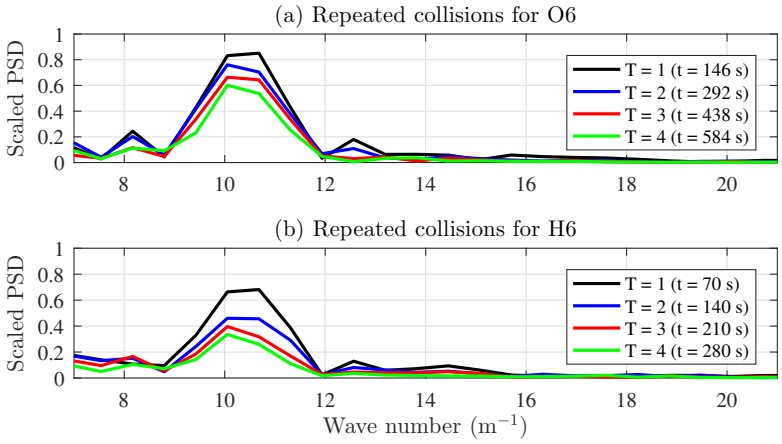

**Figure 12.** Scaled PSD of the cases (a) O6 and (b) H6 after repeated collisions.

**Table 4.** Quantitative measurement of the peak values observed in figure 12.

| Simulation | $T = 1$ | $T = 2$ | $T = 3$ | $T = 4$ |
|---|---|---|---|---|
| O6 | 85.11 % | 76.08 % | 66.40 % | 60.20 % |
| H6 | 68.18 % | 46.03 % | 39.64 % | 33.61 % |

note that this observation is consistent with the result shown in figure 9, since above the critical wavelength $\lambda = 0.5$ m, very little energy exchange occurs due to the interaction.

For the cases O6 and H6, simulations were performed for an extended period of time in order to allow for repeated collisions between the solitary wave and the linear wave packet. For each of these cases, four complete collision cycles were observed, and the PSD has been computed at $T = 1, 2, 3$ and $4$, as shown in figure 12. The corresponding measurement of the scaled PSD at each peak is given in table 4. The figure and table suggest that for both cases, the scaled PSD is reduced after each subsequent collision, down to 64.81 % in the case B6 and 35.71 % in the case H6 at $T = 4$. While these values are significantly smaller than those at $T = 1$, they are still larger than the scaled PSD of the short waves after only one collision, implying again that the wavelength is a much more important factor that determines the wave energy being transferred. The figure and table also show that after each collision cycle, the scaled PSD of the case H2 is always less than that of the case O2, implying that a head-on collision is more efficient in destroying the linear waves at this wavelength.

## 5.3 Influence of the interaction on the ISW

During the collision with the linear wave packet, the solitary wave is also affected by the linear waves that pass through it. We note however, that the kinetic energy carried by the linear waves is much smaller than that carried by the solitary wave, and hence the impact of linear waves on the solitary wave is also small. Here, we define the kinetic energy (KE) per unit mass





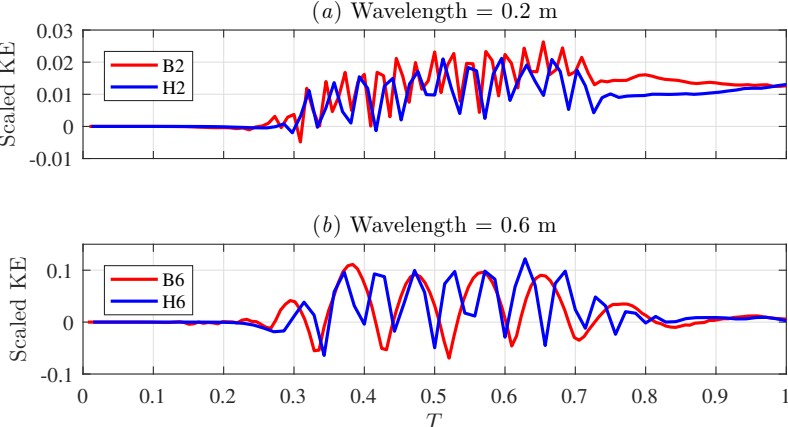

**Figure 13.** Time series of scaled maximum vertically integrated kinetic energy. The figure shows difference between simulations with and without the linear waves. Note the different scales in $y$-axis.

following standard convention (which drops the reference density and hence changes the dimensions of the quantity) by

$$\text{KE} = \frac{1}{2}(u^2 + w^2). \tag{15}$$

We found that when measured in terms of vertically integrated kinetic energy at the wave crest, the linear waves are about 1 % as energetic as the solitary wave.

To analyze changes in the solitary wave and determine if they are results of the collision, we performed an additional simulation with the same solitary wave but without the linear waves. We estimated the vertically integrated KE at the crest of the ISW for simulations with and without linear waves, and plot the difference as time series (i.e. as functions of scaled time T) in figure 13 over one complete collision cycle. Mathematically, this quantity is computed as

$$\frac{1}{A} \max_{0 \leq x \leq L_x} \left[ \int_0^{L_z} (\text{KE}_{\text{full}} - \text{KE}_{\text{isw}}) dz \right], \tag{16}$$

where $A$ is the normalization factor defined as the maximum vertically integrated KE of the initial solitary wave. The subscript *full* denotes variables from simulations with both solitary and linear waves, and the subscript *isw* denotes variables from simulations with a freely propagating solitary wave. For linear waves with a wavelength $\lambda = 0.2$ m shown in panel (a), there is a net energy transfer into the solitary wave as a result of the interaction, such that the maximum vertically integrated KE has increased by at least 1 %. We are able to confirm that such an energy increase in the solitary wave is robust since we have

also performed additional simulations with a longer linear wave packet (not shown), and found that the maximum vertically integrated KE increases approximately linearly with respect to the length of the wave packet. On the other hand, for waves with a wavelength $\lambda = 0.6$ m shown in panel (b), energy increase in the solitary waves after the collision is insignificant. In all cases, the curves shown demonstrate periodicity associated with their particular wavelengths.



We have also attempted to detect the phase shift of the solitary waves from the locations of maximum vertically integrated KE. However, we found that such a phase shift, if it exists at all, is on the order of millimeters. In other words, the detected phase shift is on the grid scale and is subject to numerical error. For this reason, the results are not shown here.

## 6 Conclusions

In this work we performed two-dimensional direct numerical simulations to study the interaction between a large-scale, fully nonlinear ISW and small-scale linear internal waves. We demonstrated that there was a net energy transfer from the small-scale linear waves to the large-scale solitary waves. This contrasts the conclusion in Broutman and Young (1986), made for a different type of internal wave interaction, that energy is transferred from large-scale waves to small-scale waves. Our simulation results suggest that during the interaction, the solitary wave essentially acts as a filter through which only long waves may pass. For waves with a smaller wavelength, the interaction leads to a reduction of their initial energy and a destruction of their spatial structure. These processes occur in a background state set by the solitary wave-induced stratification and current. During the interaction, adjustment of the short waves to this new background environment extracts their wave energy and modifies the wave structure. The fact that short waves may not survive the interaction with a solitary wave, or more generally, any localized nonlinear background environment which both deforms the pycnocline and induces shear, implies that the observed spectrum of wavelengths of internal waves in locations with large amplitude ISWs (such as Straits) is likely to be poor in short waves. At the time of writing we are unaware of measurements to support or contradict this hypothesis.

We performed analysis based on linear wave theory and showed that during the nonlinear interaction with the ISW, the destruction of short linear waves occurs primarily due to the presence of ISW-induced velocity shear, which alters the vertical structure of the short waves in a nonlinear manner, leading to significant wave amplitudes on only one side of the deformed pycnocline center. On the other hand, a shift of the location of pycnocline plays a secondary role during the collision, as its main effect is to alter the propagation speed of the linear waves, and shift the location of the maximum of the vertical structure downward. However, the vertical structure is unchanged with respect to the pycnocline center. We also demonstrated that a critical layer is not present during the collision, regardless of the wavelength of the linear waves, since the phase speed approaches the maximum ISW-induced current asymptotically as the wavelength approaches zero.

A clear avenue of future research is to explore the parameter space, in particular the Richardson number effect, of the solitary wave. In the present work we studied the ISW whose minimum Richardson number is 0.45. While this suggests that no shear instability has been generated, it also indicates that a Richardson number smaller than 0.25 is not necessary for the short waves to be destroyed due to collision with the ISW. Nevertheless, this does not mean that the wave-wave interaction considered in the present work is Richardson number independent. Moreover, Lamb and Farmer (2011) showed that it is not only the minimum Richardson number, but also the length of the unstable region with a low Richardson number relative to the wavelength of ISW, that is jointly responsible for the generation of shear instability in an ISW. It is thus reasonable to assume that the relative length of the region with a low-Richardson number in the ISW also has an influence on the wave-wave interaction. Future research will explore these effects in detail.



We note that our findings are in many ways similar to those in Stastna et al. (2015). Their study also concluded that the direction of energy transfer during the interaction is from the small-scale, weakly nonlinear wave (i.e. the mode-2 wave) to the large-scale solitary wave (i.e. the mode-1 wave), and that such energy transfer is more efficient when a head-on, instead of overtaking, collision is involved. The main difference is that in mode-1-mode-1 interaction, there is a cutoff determined by

the wavelength of short waves, above which the small-scale waves maintain their structure after interaction, whereas mode-1-mode-2 interaction is much less dependent on the wavelength. In mode-1-mode-1 interaction, this cutoff corresponds to the wavelength at which the phase speed of the short waves upstream of the solitary wave exceeds the maximum ISW-induced current. In mode-1-mode-2 interaction, however, this cutoff does not exist since the maximum ISW-induced current is always larger than the phase speed for any given wavelengths.

While all of the simulations discussed in this work are performed on the laboratory scale, the scaling-up of the current experiments to the field scale is left as a topic for future work. When the field scale is considered, waves with a much larger range of wavelengths can be expected to breakdown, including short waves affected by self-interaction (Sutherland, 2016). Also, a higher Reynolds number implies that the overturning seen in figure 5 may eventually lead to significant overturns. The three-dimensionalization of the flow field should also be examined. As shown in Andreassen et al. (1994), two-dimensional

models are unable to describe properly the physics or the consequences of the wave breaking process, in particular that induced by the presence of a critical layer. We also note that in two dimensions, the only possible form of wave-wave interaction is either an overtaking collision or a head-on collision. However, observational evidences (e.g. Quaresma et al. (2007); in particular, see their figures 2 and 8) suggest that internal waves do not generally propagate parallel to each other but may interact at different angles. The effects of directionality of wave propagation is another topic that can be considered in forthcoming studies.



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
