# Peer review of "On the interaction of short linear internal waves with internal solitary waves"

_Nonlinear Processes in Geophysics, 2017_

## Referee Comment (RC1) · Anonymous Referee #1 · 8 Oct 2017

General comments

The paper contains new results about an interaction between a large-scale, fully non-linear internal solitary waves (ISW) and small-scale linear internal waves. Resonant behavior is found for some wavelengths, when the energy is transferred from small amplitude relatively short wave into the ISW. This effect is new and is important for further understanding of properties and dynamics of fully nonlinear internal solitary waves and for practical applications.

The topic, title, abstract and text of the paper are appropriate for the Nonlinear Processes in Geophysics Journal. A good review of the problem context is given in the Introduction. All figures are necessary and of good quality, except for minor misprints (q.v. Technical Corrections). Conclusions are clear and concise.

The paper can be published after minor revision.

Specific comments

1. The arguments are made in general terms such as Âńwaves that are short in comparison to the ISW length". It would be good to give the spectrum of the solitary wave to see where are the wavenumbers of the considered small-scale monochromatic waves with respect to characteristic wavenumbers in ISW spectrum, its width, etc.

2. Particular solitary wave is considered in the paper, and the amplitude of linear waves is set to 1 mm for all cases (it is mentioned that simulations with an amplitude of 2 mm have produced quantitatively the same results). Only wavelength of the linear waves was changing. It would be interesting to tune the parameters of the problem, to investigate how they influence the process interaction and to find possible parameterization of the problem. For example, authors could try to change the width of the solitary wave (of course, together with its amplitude) keeping constant wavelength of the linear wave and saving the ratio of the length scales ("short" and "long" ISW in comparison to linear wave). Such tasks are claimed as a proposal for future research.

3. Also it is interesting to understand the amplitude-wavenumber limits of "linearity" of the "linear" waves. This would be interesting to discuss within the present paper.

4. Page 15, first paragraph: "According to the linear wave theory, the wave induced horizontal velocity is much larger than the vertical velocity" – this is true only for long waves, but linear waves considered in the paper are not proven to be long. In the theory of internal waves the criterion for waves to be long is based not only upon the ratio of wavelength and the total height of the water column (as it is for surface waves), but includes also some characteristic of the density stratification, and thus is weaker than that for surface waves, nevertheless authors should prove that their linear waves can be treated as long internal waves.

5. Authors state that "a critical layer is not present during the collision" based on the

linear criterion Ri > 0.25. But Figures 3 (at T=0.5) and 5 for O2 and H2 cases clearly demonstrate the presence of instability (at least instability of Rayleigh – Taylor), which manifests itself in an overturning, in contrast to Figure 4 for O6 case, where there is no any instability. How can this instability be explained?

Technical corrections

1. Not all variables and notations are described when the equations (1), (3) are formulated.

2. I can't see dotted horizontal lines, which are described in the annotation to Figure 6.

3. Line 7, page 17: what is the case B6, and why values of scaled PSD (in %) are not the same as in Table 4 at T=4 for both O6 and H6? Also, what are B2 and B6 in the legends in Figure 13?

---

## Editor Comment (EC1) · K. Terletska (Editor) · 12 Oct 2017

Understanding of dynamics of interaction between a fully nonlinear internal solitary waves and linear internal short waves, estimation of energy transfer during two types of interaction (head-on collision and overtaking) are of great importance for understanding of the nonlinear internal wave dynamics. The paper highlighted properties of nonlinear internal solitary waves that are new and interesting. The topic, title, abstract and text of the paper are appropriate for the NPG Journal. I also have some comments:

1. Title: "On the interaction of short linear waves with internal solitary waves" It should be mentioned that short linear waves are also internal.

2. Page.6. Could you please explain in more detail estimation of wavelength. It is not

clear.

3. Page 6. "The amplitude of linear waves is set to be 1 mm(!) for all cases." - It seems it is the mistake, too small amplitude. I guess that the amplitude of linear internal waves was 1 cm. And the vertical resolution is about 1 mm.

4. Than the question arises: Is the resolution fine enough to resolve linear internal short waves?

5. Page 7 " In fact, simulations with an amplitude of 2 mm(!) have produced quantitatively similar results, and thus will not be discussed in this paper."– Same mistake. Probably "amplitude is 2 cm"

6. Page 7. What is the value Lx in the equation (8)

7. Page 9. Fig. 4. Why ISW in this figure moves backward? In Fig. 4. (c) at t=146 s the ISW has shifted backward to x=7.2. Because on the previous frame in Fig.4. (b) ISW was located around x =9.7 at t=73 s.

8. Page 16 line 5 mistake: "the linear waves have an amplitude of 0.2 m (!) " Probably "amplitude is 2 cm"

9. I didn't find the value of amplitudes of mode-2 waves.

---

## Referee Comment (RC2) · Anonymous Referee #2 · 19 Nov 2017

This paper investigates the interaction of short linear waves with internal solitary waves numerically. The interaction mechanism and energy transport during the interaction is provided in detail. The results are new and interesting, which provided a view for the further study of nonlinear internal wave dynamic. Overall, the paper is well organized and written, including sufficient experiments and reliable discussion. The topic, title, abstract and text of the paper are appropriate for the NPG Journal, the figures are of good quality. The conclusion is reliable and concise. Some minor comments are as follows.

1. In line 26 at page 6, the author said the amplitude of linear waves is set to 1mm for all cases, however, from the Figure 3 and Figure 4, the amplitude of linear waves is obviously higher than that value.

[Figure]

2. In section 4.2, the author describes the destruction of short waves in detail. According to the Figure 5, we can find the overturning significantly. The generation of this overturning is interesting, it possibly triggered by the weakening of stratification during interaction between linear waves and ISWs. It would be better to give some discussions on its generation mechanism, some values of Richardson number or Froude number could also be provided.

3. The destruction of short linear waves is subject to a modified stratification and a velocity shear. It would be interesting to provide a more detailed discussion about the adjustment of waves caused by a modified stratification.

4. According to the section 4.4, the phase speed of linear waves was modified by the interaction. The Froude number could be introduced to analyze the nonlinearity changing during interaction.

5. In lines 12-19 at page 18, the KE of ISWs increased by at least 1% after the interaction with the linear wave with a wavelength of 0.2m, however, for waves with a wavelength of 0.6m, the KE of ISWs didn't increase significantly. More discussion of the relationship between the wavelength of linear waves and the increasing energy of ISWs should be provided since this transport process is an interesting point to the readers.

---

## Author Comment (AC1) · 30 Nov 2017

We thank the editor and referees for their time and energy put into the review. We have considered all of the comments and provide point by point discussion below. The original comments from the Editor and Referees are in black while the authors response is in blue. In addition, we also provide a summary of our changes in manuscript and attach the revised manuscript as a supplement.

Before replying to the specific comments, we would like to mention that there was a mistake in the computation of Richardson number in the original manuscript. The correct value of the minimum Richardson number should be 0.246. While this value is much smaller and indeed smaller than the critical Richardson number 0.25, it does not

change the conclusion that there is no shear instability occurring during the interaction. This is because the onset of shear instability in ISWs is a much more complicated phenomenon than that in a parallel shear flow, and is subject to multiple restrictions, as suggested in Lamb and Farmer (2011). Based on the criteria concluded in Lamb and Farmer (2011), the onset of shear instability is not likely to occur in the ISW considered in our work. We have corrected this in table 1 and changed the relevant discussion in the revised manuscript. We have also added a new Richardson number plot (figure 3) to show the values of local Richardson number of the ISW.

**Reply to the Editor**

1. Title: "On the interaction of short linear waves with internal solitary waves" It should be mentioned that short linear waves are also internal.

We have changed the title to "On the interaction of short linear internal waves with internal solitary waves" in the revised manuscript.

2. Page.6. Could you please explain in more detail estimation of wavelength. It is not clear.

The wavelength is computed according to the formula $\lambda_{isw} = 2(x_R - x_L)$, where $x_R$ and $x_L$ satisfy the equation $u(x_R, L_z) = u(x_L, L_z) = 0.5u_{\max}$. This has been added to the caption of table 1 in the revised manuscript.

3. Page 6. "The amplitude of linear waves is set to be 1 mm(!) for all cases." - It seems it is the mistake, too small amplitude. I guess that the amplitude of linear internal waves was 1 cm. And the vertical resolution is about 1 mm.

This is a typo. The amplitude should be 1 cm. We have fixed it in the revised manuscript.

4. Than the question arises: Is the resolution fine enough to resolve linear internal short waves?

From a purely numerical point of view, the spectral method requires a minimum of two points in the horizontal direction in order to completely resolve a wave (Trefethen, 2000). Thus, the nature of spectral method guarantees that the resolution is more than enough for resolving the short waves. We have explained this in the last paragraph of section 2.2 in the revised manuscript.

5. Page 7. "In fact, simulations with an amplitude of 2 mm(!) have produced quantitatively similar results, and thus will not be discussed in this paper." Same mistake. Probably "amplitude is 2 cm".

Corrected.

6. Page 7. What is the value $L_x$ in the equation (8).

$L_x = 10$ m, as stated in the first paragraph of section 3.2.

7. Page 9. Fig. 4. Why ISW in this figure moves backward? In Fig. 4. (c) at $t = 146$ s the ISW has shifted backward to $x = 7.2$. Because on the previous frame in Fig.4. (b) ISW was located around $x = 9.7$ at $t = 73$ s.

Corrected. The mistake was due to the use of periodic boundary condition.

8. Page 16 line 5 mistake: "the linear waves have an amplitude of 0.2 m (!)". Probably "amplitude is 2 cm".

Corrected.

9. I didn't find the value of amplitudes of mode-2 waves.

It is also 1 cm.

**Reply to Referee #1**

Specific comments

1. The arguments are made in general terms such as "waves that are short in comparison to the ISW length". It would be good to give the spectrum of the solitary wave to see where are the wavenumbers of the considered small-scale monochromatic waves with respect to characteristic wavenumbers in ISW spectrum, its width, etc.

We added a new figure (figure 12) to demonstrate the spectra of the horizontal velocity of the short waves and the solitary wave.

2. Particular solitary wave is considered in the paper, and the amplitude of linear waves is set to 1 mm for all cases (it is mentioned that simulations with an amplitude of 2 mm have produced quantitatively the same results). Only wavelength of the linear waves was changing. It would be interesting to tune the parameters of the problem, to investigate how they influence the process interaction and to find possible parameterization of the problem. For example, authors could try to change the width of the solitary wave (of course, together with its amplitude) keeping constant wavelength of the linear wave and saving the ratio of the length scales ("short" and "long" ISW in comparison to linear wave). Such tasks are claimed as a proposal for future research.

We decided to focus on the effects of wavelengths on the interaction in this work. This is because a change in the width of the solitary wave does not only change the wavelength of the linear waves relative to the solitary wave, but also changes a number of other parameters such as the Reynolds number and the Richardson number, all of which are likely to have some effects on the interaction (Lamb and Farmer (2011) discussed some of the complications in detail). This topic is too broad for the present work but will probably be picked up in a future publication.

3. Also it is interesting to understand the amplitude-wavenumber limits of "linearity" of the "linear" waves. This would be interesting to discuss within the present paper.

We added a paragraph in section 2 to discuss the KdV theory and based on the KdV equation, we estimated the nonlinear time scale of the "linear" waves to be order of 1000 seconds (last paragraph in section 3.2). This is much larger than the time scale of interaction, and thus the short waves can indeed be considered "linear" in our simulations.

4. Page 15, first paragraph: "According to the linear wave theory, the wave induced horizontal velocity is much larger than the vertical velocity" ? this is true only for long waves, but linear waves considered in the paper are not proven to be long. In the theory of internal waves the criterion for waves to be long is based not only upon the ratio of wavelength and the total height of the water column (as it is for surface waves), but includes also some characteristic of the density stratification, and thus is weaker than that for surface waves, nevertheless authors should prove that their linear waves can be treated as long internal waves.

We removed this sentence since, as the referee pointed out, the waves considered in our study are not proven to be long. It is true that in this case the horizontal velocity does not dominate the wave energy. However, the point of the PSD computation is to measure the reduction of wave energy (i.e. the ratio of the wave energy between the final and initial fields) instead of the wave energy it self. Since computation of the PSD of the vertical velocity field yields quantitatively similar results, whether using the horizontal or vertical velocity, or even kinetic energy, for the PSD computation does not change our conclusions. We have added a sentence in the last paragraph of section 5.1 to explain this.

5. Authors state that "a critical layer is not present during the collision" based on the linear criterion $> 0.25$. But Figures 3 (at $T = 0.5$) and 5 for O2 and H2 cases clearly demonstrate the presence of instability (at least instability of Rayleigh-Taylor), which manifests itself in an overturning, in contrast to Figure 4 for O6 case, where there is no any instability. How can this instability be explained?

We concluded that "a critical layer is not present" based on the computation of phase speed discussed in section 4.4, and that shear instability is not likely to occur based on the criteria concluded in Lamb and Farmer (2011). The overturning observed in the cases O2 and H2 is due to focusing of the energy of linear waves in the adjusted structure as predicted by linear theory (figure 7 in the revised manuscript). This focusing of wave energy occurs either above (for overtaking collisions) or below (for head-on collisions) the pycnocline. Detailed discussion is given in section 4.2.

Technical corrections

1. Not all variables and notations are described when the equations (1), (3) are formulated.

The definitions of $\nabla$ and $N$ have been added.

2. I can't see dotted horizontal lines, which are described in the annotation to Figure 6.

We removed the relevant sentence in the figure caption. From the structure functions the location of the undisturbed pycnocline center should be clear.

3. Line 7, page 17: what is the case B6, and why values of scaled PSD (in %) are not the same as in Table 4 at $T = 4$ for both O6 and H6? Also, what are B2 and B6 in the legends in Figure 13?

The case labels "B2" and "B6" should be "O2" and "O6". We have corrected these notations. The values shown in table 4 should be correct, and we have corrected the corresponding values in the text.

**Reply to Referee #2**

1. In line 26 at page 6, the author said the amplitude of linear waves is set to 1 mm for all cases, however, from the Figure 3 and Figure 4, the amplitude of linear waves is obviously higher than that value.

This is a typo and the amplitude should be 1 cm. We have fixed it in the revised manuscript.

2. In section 4.2, the author describes the destruction of short waves in detail. According to the Figure 5, we can find the overturning significantly. The generation of

this overturning is interesting, it possibly triggered by the weakening of stratification during interaction between linear waves and ISWs. It would be better to give some discussions on its generation mechanism, some values of Richardson number or Froude number could also be provided.

The overturning is due to focusing of the energy of linear waves in the adjusted structure as predicted by linear theory in figure 7 (in the revised manuscript). This focusing of wave energy occurs either above (for overtaking collisions) or below (for head-on collisions) the pycnocline. It is true that the stratification weakens on either side of the pycnocline, but so does the velocity shear. In fact for the ISW considered in our simulations, the Richardson number is at its minimum inside the density interface along the wave crest but is very large outside of the density interface. Therefore, the overturning is not triggered by the weakening of the stratification. We have added a new figure to show the local Richardson number (figure 3) in the revised manuscript. For the estimated Froude number, please see reply to comment #4.

3. The destruction of short linear waves is subject to a modified stratification and a velocity shear. It would be interesting to provide a more detailed discussion about the adjustment of waves caused by a modified stratification.

For fully nonlinear ISWs, the isopycnal displacement and wave-induced current are set by the particular solutions of the DJL equation and cannot be varied arbitrarily. This means a change in the isopycnal displacement will necessarily lead to a change in the wave-induced current, and hence the velocity shear. For this reason, it is difficult to examine the adjustment of linear waves to a modified stratification and a velocity shear separately, because the interaction is indeed a nonlinear process.

4. According to the section 4.4, the phase speed of linear waves was modified by the interaction. The Froude number could be introduced to analyze the nonlinearity changing during interaction.

In the context of internal wave dynamics, the Froude number is usually defined as

$$Fr = \frac{U}{c},\qquad(1)$$

where $U$ is the background current and $c$ is the linear wave speed. For linear waves away from the ISW, the Froude number is essentially zero since there is no background current. For linear waves within the ISW when interaction occurs, the vertically integrated $U$ is also zero since the flow is non-divergent in 2D. Thus, a better estimation of $U$ would be the effective horizontal velocity in a reference frame moving with the ISW, which is essentially $-c_{isw}$. In this reference frame, the estimated $c$ would be $-c_{isw} + c_p$ where $c_p > 0$ for an overtaking collision and $c_p < 0$ for a head-on collision. Thus, in the revised manuscript we define

$$Fr = \frac{c_{isw}}{c_{isw} - c_p},\qquad(2)$$

where $Fr < 1$ for an overtaking collision and $Fr > 1$ for a head-on collision. In both cases, $Fr$ is close to 1 only for short waves since they propagate slower, implying that nonlinear effects are more important in the interaction of ISWs with short waves. We have added a new figure (figure 11) to show the Froude number as a function of wavelength in the revised manuscript.

5. In lines 12-19 at page 18, the KE of ISWs increased by at least 1 % after the interaction with the linear wave with a wavelength of 0.2 m, however, for waves with a wavelength of 0.6 m, the KE of ISWs didn't increase significantly. More discussion of the relationship between the wavelength of linear waves and the increasing energy of ISWs should be provided since this transport process is an interesting point to the readers.

In fact, only when the waves are destroyed do we get modification of ISW energy. For interaction of ISWs with longer waves, the PSD plots showed that there is very little loss of linear wave energy after the interaction. Thus, it is highly unlikely that the KE of the ISW will increase significantly.

**Authors' changes in manuscript**

Other than correcting all the typos, we have made several major changes in the revised manuscript. These changes are marked up in blue.

1. Page 1. We changed the title to "On the interaction of short linear internal waves with internal solitary waves".

2. Page 4. We added a paragraph summarizing the weakly nonlinear theory. We also added a description of the numerical method used for solving the DJL equation.

3. Page 5. We added a sentence to describe the high accuracy nature of the spectral method.

4. Page 7. 1) We corrected the minimum Richardson number in table 1. 2) We added the formula for computing the wavelength in table 1. 3) We added figure 3 to show the horizontal velocity of the ISW and contours of local Richardson number. 4) We modified the relevant discussion based on the corrected value and the new figure.

5. Page 9. We added a paragraph to discuss the nonlinear time scale of the short waves.

6. Page 15. 1) We changed figure 10 to show the phase speed as a function of wavelength for both overtaking and head-on collisions. 2) We added figure 11 to show the Froude number as a function of wavelength.

7. Page 16. We added a paragraph to discuss the calculation and interpretation of the Froude number

8. Page 17. We added figure 12 to show the spectra of the ISW and the linear waves in the initial field. We also added some discussions relevant to this figure.

**Supplement:**

[revised manuscript text omitted]